# Bioaerogels: Promising Nanostructured Materials in Fluid Management, Healing and Regeneration of Wounds

**DOI:** 10.3390/molecules26133834

**Published:** 2021-06-23

**Authors:** Beatriz G. Bernardes, Pasquale Del Gaudio, Paulo Alves, Raquel Costa, Carlos A. García-Gonzaléz, Ana Leite Oliveira

**Affiliations:** 1Universidade Católica Portuguesa, CBQF-Centro de Biotecnologia e Química Fina–Laboratório Associado, Escola Superior de Biotecnologia, Rua Diogo Botelho 1327, 4169-005 Porto, Portugal; bbernardes@porto.ucp.pt; 2Department of Pharmacology, Pharmacy and Pharmaceutical Technology, I+D Farma Group (GI-1645), Faculty of Pharmacy and Health Research Institute of Santiago de Compostela (IDIS), Universidade de Santiago de Compostela, E-15782 Santiago de Compostela, Spain; 3Department of Pharmacy, University of Salerno, Via Giovanni Paolo II 132, I-84084 Fisciano, Italy; pdelgaudio@unisa.it; 4Center for Interdisciplinary Research in Health, Institute of Health Sciences, Universidade Católica Portuguesa, Rua Diogo Botelho 1327, 4169-005 Porto, Portugal; pjalves@porto.ucp.pt; 5Instituto de Investigação e Inovação em Saúde, Universidade do Porto (i3S), 4200-135 Porto, Portugal; 6Biochemistry Unit, Department of Biomedicine, Faculdade de Medicina, Universidade do Porto, 4200-319 Porto, Portugal; 7Escola Superior de Saúde, Instituto Politécnico do Porto, 4200-072 Porto, Portugal

**Keywords:** aerogels, biopolymer, exudate, wound healing

## Abstract

Wounds affect one’s quality of life and should be managed on a patient-specific approach, based on the particular healing phase and wound condition. During wound healing, exudate is produced as a natural response towards healing. However, excessive production can be detrimental, representing a challenge for wound management. The design and development of new healing devices and therapeutics with improved performance is a constant demand from the healthcare services. Aerogels can combine high porosity and low density with the adequate fluid interaction and drug loading capacity, to establish hemostasis and promote the healing and regeneration of exudative and chronic wounds. Bio-based aerogels, i.e., those produced from natural polymers, are particularly attractive since they encompass their intrinsic chemical properties and the physical features of their nanostructure. In this work, the emerging research on aerogels for wound treatment is reviewed for the first time. The current scenario and the opportunities provided by aerogels in the form of films, membranes and particles are identified to face current unmet demands in fluid managing and wound healing and regeneration.

## 1. Introduction

Wound healing involves complex cellular, biochemical and physiological events which are dependent of a myriad of local, systemic and environmental factors, including wound humidity, oxygenation, infection and maceration, as well as age, nutritional status, obesity, medications and smoking habits [1,2,3]. After the healing process, the injured site should have connective tissue repaired and re-epithelialized and returned to its normal anatomical structure and functional integrity [1]. Wounds can be categorized into acute or chronic according to their healing process mechanism. Acute wounds occur due to the loss of skin integrity after a burn, trauma or surgery event and follows a regular healing process within 3 weeks. Wounds with an abnormal healing process of a minimum of 3 months are known as chronic wounds [2,4,5]. Chronic wounds can be classified according to their etiology, wound location and appearance (deep, shallow, eschar and exudative) being designated as venous ulcer, pressure ulcer, diabetic foot ulcer or arterial ulcer [6,7,8].

The skin wound healing process usually occurs in a cascade of coordinated and systemic events with a variety of cellular activities and characterized by a set of consecutive and overlapped phases: hemostasis, inflammation, proliferation/maturation/remodeling [1,9]. The transition between these stages usually depends on the maturation and the differentiation of the most important cells, namely macrophages, mast cells, fibroblasts and keratinocytes and the activation of endothelial cells [10,11,12]. The healing of chronic wounds is particularly prolonged in the inflammatory phase with insufficient anatomic and functional integrity to reach the proliferation phase [2,13]. The microflora of chronic wounds is usually polymicrobial, i.e., with the presence of several bacterial species, and these microbial infections can adversely affect the wound healing process [14]. During infection, inflammatory cells and mast cells are attracted, fluid accumulates at the wound area and inflammation is exacerbated, which adversely impact wound healing due to the increased presence of free radicals, proteases, and other bioactive substances that delay skin regeneration [15].

Chronic wounds represent approximately 3% of total healthcare expenditure in developed countries [16]. According to the European Association for Injury Prevention and Safety Promotion (EuroSafe), the leading causes of injury in the European region, excluding surgical/medical complications, between 2012–2014, were falls (50%), followed by traffic accidents (12%), and cuts/piercings and poisonings resulting in a similar magnitude of hospital admissions (5 and 4%, respectively), while burns/scalds accounted for approximately 1% of admissions [17]. Other injuries were caused by poisoning or by unknown facts. Surgical wounds, when infected or opened, may result in a chronic wound [18]. The increasing prevalence of non-communicable diseases (chronic diseases) and the ageing population have put the spotlight on wound care and the considerable societal burden of wounds [19]. Chronic wound healing is one of the major therapeutic and economic challenges to healthcare services. Understanding and overcoming the factors/variables that contribute to delaying or disrupting the healing process are key components for a comprehensive approach to wound care and present one of the main challenges to chronic wound management and exudate management. An accurate diagnostic procedure with a complete patient history together with an individualized wound treatment and management strategy with novel therapeutic strategies, such as wound dressings, are urgently needed to decrease wound prevalence, comorbidities and medicare costs [6,20].

Nowadays, the design and development of new biocompatible and biodegradable materials for wound management that induce and promote the healing process—repairing damaged tissue and restoring its integrity—are a constant need [3]. A healing device should be designed to promote tissue repair, exudate removal and prevent infections at the wound site [13]. Moreover, the local administration of antibiotics at therapeutic concentrations can be advantageous to reduce wound bioburden, avoiding infection scattering and systemic toxicity through other administration routes (e.g., oral delivery) [21]. Wound dressings provide a physical barrier to the wound environment against external contaminations and can incorporate substances with local antibacterial properties that could help wounds to heal in a faster way [13]. There is a diversity of dressings commercially available for exudative wound treatment, but many still lack adequate absorption kinetics and fluid orientation that may cause tissue maceration and discomfort to the patient.

Aerogels are a special class of porous material studied as components of wound dressings due to their unique properties such as high porosity, low bulk density, high surface area, ultra-light and high water uptake [22,23]. Bio-based aerogels or bioaerogels, i.e., aerogels from natural polymer (polysaccharides, protein) sources, have received special attention for biomedical applications due their biocompatibility, biodegradability and abundance [22,24]. Research on aerogels for wound treatments only started in the 2010s and is still in an incipient status, although with a clear increasing trend of the number of articles published per year in the literature (Figure 1) and of research groups and collaborative projects focused on this topic worldwide [25]. Previous review articles on aerogels are either focused on processing methods sorted by the polymer source of the aerogel [26,27,28], or on biomedical applications in general [22,29,30,31] without stressing the specific needs for wound healing applications. To the best of our knowledge, this review article describes for the first time the developed technologies for the synthesis and processing of bio-based aerogels in the form of films, membranes and particles, specifically intended for the treatment of chronic and exudative wounds. Based on the current clinical demands for wound treatment, a critical discussion on the advantages of bio-based aerogels for each wound healing stage and for exudate control to accelerate wound healing will be carried out.

## 2. Challenges and Opportunities to Heal Exudative Wounds

During wound healing, a fluid can be produced in the inflammatory and proliferative phases as a natural response towards healing. The so-called exudate is a physiological and essential part of the wound healing process promoting healing and a moist wound environment [32,33]. Exudate is mainly composed of water and a cocktail of elements including nutrients, electrolytes, inflammatory mediators, proteins, proteolytic enzymes, growth factors, various cells types and waste products [34,35]. The presence of proteins in high contents in the exudate is essential as a nutrient for epithelial cells and enables the ingress of white cells. Electrolytes and inflammatory components (leukocytes, fibrinogen and fibrin) in the exudate provide the optimal environment to promote cell proliferation and autolysis [32,33]. Besides the aforementioned benefits, if there is a continuous and excessive production of exudate in the wound bed, it could be detrimental for the healing process as it prolongs the inflammatory response, is a source of embarrassment for the patient and represents a challenge for clinical wound management [32,33].

### 2.1. Wounds Stages

In the event of a wound, the body should adapt to it and deploy a series of biochemical reactions to repair the injured tissues. This series of reactions involves the local synchronization of a variety of cell types, growth factors, and is divided into a set of consecutive and overlapping stages (Figure 2): hemostasis, inflammatory, proliferative and maturation [36].

#### 2.1.1. Haemostasias

Haemostasias starts with the beginning of the injury and aims to stop bleeding. At this stage, the body activates the blood clotting cascade, its emergency repair system, and forms a dam to block drainage [37]. Haemostasias initiates the coagulation cascade, vasoconstriction and complement cascades, resulting in platelet activation, adhesion and aggregation, forming a clot that limits the blood loss and provides a provisional matrix for cell migration [1,2,3]. An enzyme named thrombin initiates the formation of a fibrin mesh, which strengthens platelet clusters in a stable clot [37]. The platelets trapped in the clot release alpha granules when exposed to collagen, which secrete several growth factors and cytokines into the local wound environment, activating fibroblasts, endothelial cells and macrophages that will act as a reservoir in the provisional matrix and will be ulteriorly released during the succeeding healing phases [1,2,38].

#### 2.1.2. Inflammation

The inflammatory stage focuses on the destruction of bacteria and removal of debris to prepare the wound bed for the growth of new tissue. The inflammatory phase is responsible for the recruitment of the innate immune system (monocytes and neutrophils), the wound cleaning by stripping dead tissue, and the beginning of rebuilding through vasodilation and macrophage activation [3,9]. This stage usually lasts four to six days and is usually associated with edema, erythema (redness of the skin), heat and pain [37]. One of the principle signs of inflammation is swelling at the wound site in consequence of the presence of fluid accumulation (exudate) [4].

The inflammatory phase begins with the infiltration of neutrophils granulocytes at the wound site, and with the phagocytose of bacteria, cell debris and other particles [1,38]. Inflammatory cells secrete pro-inflammatory cytokines, produce proteases that are present in the exudate and induce the production of growth factors that are extremely important to support new tissue formation [38,39]. Roughly 3 days after injury, monocytes are recruited to the wound environment, where they differentiate into macrophages and support healing by promoting vessel formation [9,40]. Macrophages are fundamental for the coordination of the later events during healing process. They regulate the proteolytic destruction of damaged extracellular matrix (ECM) proteins through the release of protease inhibitors, helping in the wound debridement [1,4]. As the white blood cells break, the macrophages arrive to continue cleaning the waste, such as the main phagocytic cells. These cells also secrete growth factors and proteins that attract immune cells to the wound to facilitate tissue repair [38].

Prolonged inflammation is unfavorable, delaying the normal progress of wound healing, being responsible for excessive scarring and possibly leading to non-healing wounds [7,9,38]. In chronic wounds, the inflammation phase is prolonged and increased due to poor vascularization and consequent hypoxia, and the inability of immune cells to control bacterial infection.

#### 2.1.3. Proliferation/Maturation/Remodeling

Once the wound is clean, the healing process begins the proliferative stage, where the goal is to fill and cover the wound; the wound contracts as the new skin tissue is built. The proliferative stage has three distinct phases: wound filling; the contraction of the wound margins; and the covering of the wound (epithelialization). During this stage, dark, shiny red granulation tissue fills the wound bed with connective tissue and new blood vessels are formed.

The proliferative phase can take weeks after wound injury and is characterized by fibroblast and keratinocyte migration, angiogenesis, the deposition of ECM and epithelialization [1,3]. The epithelial cell proliferation starts with chemotactic stimulation with increased microvascular permeability that allows, through the leakage of proteins, cytokines and growth factors, the formation of a provisional fibrin ECM in the wound site, which is necessary for angiogenesis [1,9,38]. The growth of new blood vessels is a key event during this phase, being responsible for replacing the previously damaged vessels and restoring circulation at the wound site [2,40]. Additionally, other skin cells, namely keratinocytes and activated fibroblasts, migrate from the wound edge to the wound site where they proliferate and assist in the construction of a new matrix.

Finally, in the maturation and remodeling stage, the neotissue slowly gains strength and flexibility and the main goal is the restoration of a normal skin appearance and functionality [41]. During this stage, the ECM components reorganize and mature, and there is a general increase in tensile strength [41]. The ECM is constantly remodeled to restore the normal architecture of the dermis, through the synthesis and degradation of collagen fibers type III, by the action of metalloproteinases [42]. Then, a new matrix made of densely packed collagen fibers type I is created, and most fibroblasts and inflammatory cells disappear from the wound site, forming scar tissue in the presence of growth factors [1,40]. Over time, the number of newly formed capillaries regress to give rise to a mature vessel network with a vascular density similar to that of normal skin [43].

### 2.2. Current Needs Regarding Exudative Wounds

The presence of exudate in wounds could be beneficial or harmful for wound healing depending on the wound tissue type (Table 1). The exudate management aims at: protecting the surrounding skin; optimizing the wound bed moisture level as appropriate for the patient; managing symptoms and improving the patient’s quality of life.

The exudate from acute wounds is beneficial to wound healing, and is rich in leukocytes and nutrients, stimulating cell production and proliferation [33,47]. However, the exudate present in chronic wounds possesses increased levels of proteases and contains active degrading enzymes, which adversely affect wound healing as it slows down or even blocks cell proliferation, thus interfering with growth factors, inactivates essential matrix material and damage healthy skin surrounding the wound bed [32,33,47].

Exudate is an excellent culture medium for bacterial growth that contributes to wound odor [47]. A sudden increase in wound exudate and pain are indicative of wound infection, and infection should be treated in line with local policies on the use of topical antimicrobials [48].

Most chronic wounds contain biofilm, which can disrupt healing by inducing and prolonging the inflammatory phase in the wound [49,50]. Biofilm comprises microorganisms embedded in a matrix of molecules, including proteins and DNA, that can exist on the surface of a wound and within deeper tissues and may be present as patches or islands. Even though slough is not biofilm, it is likely to contain microorganisms and may be a product of biofilm-induced inflammation [50,51,52]. Currently, there is no straightforward test for the detection of biofilms in wounds. In the absence of overt wound infection, clinical indicators of biofilm interference with the healing process include delayed healing despite optimal management, increased wound exudate levels, the failure of response to antimicrobial therapy, cycles of recurrent infection, low-level erythema and low-level inflammation [53].

### 2.3. Different Types of Exudate

Different wounds in various conditions result in varying exudate types (Table 2) [32,54]. Exudate type, color, odor and consistency can provide useful indicators of the stage of the healing process and the occurrence of possible problems that impairs wound healing [44].

The interaction between the wound dressing and the exudate type in wound site influences the wound treatment. The appropriate wound dressing should create and maintain the perfect environment for healing according to the wound status (Table 3) [58].

### 2.4. Limitations of the Current Wound Dressing Approaches for Exudative Wounds

Successful wound management requires that the healthcare professional responsible for wound care must identify and treat the underlying causes of abnormal wound exudate levels and remove excess exudate from the wound bed with the correct selection of dressings.

The understanding of the wound exudate interaction with the dressings and its ideal time of use are essential for patients’ care with highly exudative wounds. This will help prevent complications such as skin reactions and maceration caused by inadequate dressing selection and poorly timed dressing changes [60]. Educating patients and health professionals about the cause of the wound and contributory factors, the rationale for treatment and when/how to seek help if problems arise, are fundamental to shared decision making and the promotion of consensus (Table 4) [48].

Different wound dressings are used for the treatment of wounds and can be classified based on different factors, according to the wound dressing type (traditional, biomaterial-based and artificial dressings), application category (primary or secondary dressing) and interaction with biological tissue (passive and bioactive dressings) [61]. Nowadays, the dressings are made of different types of materials and forms, such as hydrocolloids, hydrogels, sponges, films, nanofibers and polymerized membranes [61,62]. In addition, dressings with antimicrobial, biological-based, anti-inflammatory and antiseptic properties, tissue-engineered skin substitutes, dressings containing naturally derived wound agents and oxygen-related dressings have been developed [62,63].

There is a diversity of dressings commercially available, but several challenges still remain in tackling the problems associated with chronic wounds, such as the fact that advanced developed dressings do not always address the problems encountered in chronic wounds for every single patient; therefore, the combination of different dressings is necessary [63]. No currently developed dressing solves the problems associated with chronic wounds, including pain, inflammation, odor, the control of bacterial infections, the induction of cell migration and proliferation, skin regeneration and the associated costs [63]. The development of a dressing that encompasses all these factors is important, given the many phases of wound healing and the differences in complications observed in different patients.

## 3. Aerogel Technology

Aerogel structures encompass a series of unique properties that are of utmost importance for wound healing applications (Figure 3A). In 2007, IUPAC coined a broad definition of aerogels comprising any microporous solid with a gaseous dispersed phase [64]. Nowadays, there is no consensus with this definition among the aerogel scientific community, since any microporous material (e.g., zeolites or MOFs) as well as sponges or foams of micron-sized pores may wrongly fall within the domain of this definition. Accordingly, the term aerogel is usually restricted to solids of low envelope density with a high open porosity (90–99.99%), mainly in the mesoporous range [23,65,66,67]. Moreover, aerogels are obtained from a wet gel precursor through a processing technique (usually supercritical drying) able to remove the pore liquid with very low structural modification. In this review article, discussion and examples are precluded to this last constrained definition of aerogels.

The physicochemical properties of aerogels have a significant role in the properties of these advanced materials for wound purposes. The dry form of the aerogels favors a rapid absorption of the wound fluid and a high shapeability of aerogels upon wetting, able to adapt to the wound morphology and topography [21,68,69,70]. The high and open porosity of aerogels is of high relevance for wound healing applications since it allows a correct gas permeability to avoid hypoxia episodes as well as a balanced transpiration-evaporation equilibrium at the wound site. This porosity also allows for a high liquid absorption capacity of certain aerogel matrices (up to 110 times its weight [71]), which is especially relevant for wound applications, since it translates in an excellent capacity to absorb exudate fluid and to manage the moisture balance at the application site [72,73,74,75]. Interestingly, this absorption capacity can be regulated by the cation concentration in the case of aerogels obtained from ionotropic gels [74,76]. Finally, the high surface area confers the aerogels with high loading capacity for bioactive agents in the pores through in-process or post-processing methods [77]. In the case of the loading of drugs, the prevalent mesoporosity of aerogels with reduced pore sizes (2–50 nm) also hampers drug crystal growth within the pores and favors the stabilization of the drug in the amorphous form favoring the dissolution of low water-soluble compounds [78].

The variety of sources and chemical functionalities for aerogel processing allow one to select nanostructured material formulations with high cell and tissue tolerability. Moreover, this portfolio of chemical groups within the aerogel structures allows one to tune the affinity of bioactive agents (e.g., drugs, enzymes, metal cations) to these matrices [72] Different aerogel formats (films [71,79], particles [73,80,81], core–shell or coated beads [82,83], fibers [84], 3D-designs [85,86], discs [87]) are under development to render high-performance aerogel-based and aerogel-containing products for wound applications. A precise selection of this combination of components and formats in the aerogel-based system will allow a customized and local release of the bioactive agent cargo ranging from fast dissolution to sustained profiles [88,89].

Silica aerogels were the first aerogel source tested for biomedical applications due to the fast development of the knowledge on their synthesis and production for other purposes (building materials and aerospace) [90,91]. Nevertheless, the high variability of biodegradation rate and degradation products of silica prompted the aerogel research toward alternative sources. The use of natural polymer-based aerogels has thus resulted in the mainstream alternative for biomedical applications in the last decade [23]. The biocompatibility, the abundance (low cost), natural availability and the body tolerance of polymers such as polysaccharides (e.g., alginate, chitosan, cellulose) or proteins (e.g., collagen, silk fibroin) are among the properties that encouraged the advent of this second generation of aerogels. Moreover, many of these natural polymers have inherent wound healing properties (e.g., alginate with high mannuronic content stimulates cytokine production [92], chitosan with antimicrobial properties [93] or cellulose–collagen composites inhibiting protease activity [94]). In a step-forward, these natural polymer-based aerogels are often being formulated in the form of composite aerogels or with the incorporation of bioactive agents (e.g., antimicrobial, anti-inflammatory, anti-proteolytic, hemostatic) with the purpose of providing additional features to the material to promote wound healing or to prevent or solve adjacent biological processes such as infections [71,73,82,94,95].

Macroporous and mesoporous networks in bioaerogels are able to promote cell migration and proliferation as well as the diffusion of nutrients into their structure boosting their use in wound care [96]. Such a network is dependent on the formation and nature of the porous arrangement formed during the gelation process [97] and the 3D structure of the gel [98]. The size and shape of bioaerogel also play an important role in the interaction with wound fluids, whereas conformability and surface area are essential to promote the insertion and adhesion into a wound cavity. In the last few years, several bioaerogel structures from different natural polymers or their composites have been investigated to serve as new promising devices (dressing films, monoliths, scaffolds, particles) for the care of various types of wounds.

### 3.1. Dressing Films

Films and sheets are the most common devices used to treat wounds (Figure 3B). The volume and type of exudate significantly determine their selection as primary and secondary dressings [99,100]. Polysaccharides characterized by abundant reserves, low prices, versatile chemical structure are ideal candidates to produce aerogel films and sheets intended for wound care, in which surface area and porosity are the main characteristics responsible for the dressing change frequency and the efficacy of wound exudate management. Sheets loaded with different bioactive agents have been proposed as primary dressings able to control the release rate of their cargo at the wound site. Impregnation of the active molecules is mainly conducted in a multistep process where the entrapment yield is determined by the film and pore diffusion of the biomolecule into the aerogel matrix. Active biomolecules have been loaded by impregnation in aerogel films based on alginate containing high mannuronic content able to induce cytokine production by human monocytes [101]. Alginate is able to form strong chemical interactions with both small molecules and mixtures of biomolecules such as mesoglycans and, when loaded in alginate aerogel films, can be released at the wound site over 24 h, enhancing its pro-migratory effect enabling the faster regeneration of the epithelium [102]. Low molecular weight chitosan, able to induce macrophages activation and to accelerate wound healing, have also been used to produce bioaerogel films and sheets [103]. The chemical structure of chitosan allows ionic crosslinking with various active biomolecules that can lead to bioaerogels with a complex pore structure, different thermal properties and proper swelling behavior when in contact with wound exudates [104]. Various cellulose hydrogels in the form of fibers have been developed for wound care applications [105,106]. Compared to aerogels obtained with conventional cellulose fibers, cellulose nanofibers aerogels have demonstrated higher porosity and the most suitable mechanical properties to be used as wound dressings [107]. Due to very high porosity and surface area, cellulose nanofiber aerogels provide good oxygen permeability, thus preventing the formation of an anaerobic environment, and can be easily loaded with an antibacterial ingredient to inhibit aerobic bacterial growth or other active compounds that can be released in a controlled manner according to the size of the nanofibers into the matrix. Natural active ingredients as well as anti-inflammatory drugs have been also incorporated into cellulose matrices. Different release profiles to the wound bed, either immediate or sustained release, can be tailored using nanofibrillar celluloses obtained from different sources, even though the cellulose materials can be chemically fairly similar [108].

Primary dressings in the form of films and sheets have also been developed from polysaccharide blends or composites [99]. Due to their functional groups, several polysaccharides can be easily blended with other biopolymers to form cross-linked network composite structures able to increase the physical stability of the wound dressing material and also provide a moist wound environment retaining a proper amount of exudate [94]. Due to its cationic moiety, chitosan is one of the most common polysaccharides used for the preparation of blended bioaerogels intended for wound care. In the case of fibers from alginate and chitosan blends, the textural properties of the aerogels (specific surface area and pore volume) were strongly dependent on chitosan molecular weight and the polymers ratio [109]. Moreover, dressings made by alginate–chitosan blends showed good antimicrobial activity against bacteria representative of wound infectious microbiota even when chitosan was present in small amounts (less than 2% (*w*/*w*)). The porosity of the blended material could be tailored by adding a proper amount of chitosan to the alginate, whereas in vitro/in vivo wound healing properties were well preserved using either low or medium molecular weight chitosan [84]. Finally, chitosan–cellulose composites in the form of films showed better pore size distribution for fluid absorption and less adhesiveness at the wound site compared to the pure cellulose [110].

### 3.2. Bioaerogel Scaffolds

Penetrating wounds resulting from chronic ulcers or debridement procedures on strongly damaged tissues (e.g., high grade burnings, diabetic foot ulcers) need tissue replacement with 3D-scaffolds or matrices, which can support cell attachment, proliferation, and subsequent wound healing. The application of biopolymer-based aerogel scaffolds directly inside the wound cavity have been proved to promote tissue growth and regeneration, acting as a template to support cellular activity (Figure 3C) [23].

Wood cellulose-based aerogel scaffolds with porosity up to 99.8% were prepared by tuning the nanocellulosic charge density through the neutralization of the electrostatic repulsion between the nanofibrils via pH assembling medium. Cells were incorporated in the cellulose aerogel scaffold during the swelling process of the gel. The reduction in the surface charge of the matrix resulted in a higher cell biological response and higher growth cell rate [111]. Cellulose-based aerogel scaffolds can be obtained by both supercritical drying treatment as well as a freeze-drying curing process [87]. A combined technique based on emulsion and freeze drying were used to fabricate a bilayer porous scaffold with two different porosities and pore sizes with an integrated layer structure. The integration between the two layers with different pore sizes allowed cell penetration in one of the layers (i.e., the fibroblast cells) and the production of ECM all over the layer, whereas the second layer acted as support for keratinocyte cells, while allowing the diffusion of growth factors and nutrients through the whole of the scaffold.

Scaffolds from cellulose nanofibers can also be printed using direct-ink-write into customizable 3D structures [112]. Highly deformable and shape recoverable cellulose nanofiber aerogels can be obtained as 3D-printed parts with low density (0.026 g/cm^3^) and high porosity (around 98%) due to the presence of polyhedron-like pore sizes in the macroporous range. Porosity, as well as shape retention after the drying process, were mainly controlled by nanofiber concentration and the viscosity of the 3D-printed cellulose solution. The 3D network of cellulose aerogels can be reinforced to increase their mechanical strength by chemical surface modification or cross-linking. The incorporation of a secondary polymer into the cellulose matrix can be obtained by supercritical carbon dioxide anti-solvent precipitation and simultaneous extraction from organogels producing the reinforced aerogel in a single process [113].

Several other biopolymers can be used to produce aerogel scaffolds with proper mechanical and moisture regulation properties for wound care applications. Alginates are particularly interesting due to their high exudate absorbency and swelling behavior. Alginate aerogel scaffolds were loaded with both biomolecules and metal cations as well as several nanoparticles releasing their cargo in human and animal injury models in a controlled manner [72,108,114]. Chitosan was combined with alginate or other biopolymers in order to produce scaffolds with properties tailored to specific target tissue and purposes. Chitosan sponges grafted with collagen peptides presented good biological activities, including the promotion of cell viability and fibroblast migration, both in vitro and in vivo (scald on rabbits), resulting as promising agents for burn care [115].

### 3.3. Bioaerogel Particles

Several recent studies have demonstrated that bioaerogel in the form of particles or beads can be successfully used in wound care (Figure 3D). In fact, they can be easily handled and introduced directly, even into deep wound cavities, by direct spreading or using devices (typically syringes); reduced size increases the instant contact and promotes the absorption of exudates in different types of wounds [26].

Particles can be produced by different dripping and atomization technologies [68,116,117] depending on the desired size and internal structure. Superabsorbent alginate aerogel beads able to absorb more saline solution than distilled water, i.e., the opposite behavior compared to regular superabsorbents, were prepared using different cross-linking ions and alginate molecular weights to tailor the final structural end properties of aerogels, thus tuning its absorption capacity and biodegradation [74]. Bioaerogels were prepared in the form of core–shell particles loaded with an antimicrobial agent, consisting of a core made by low methoxyl amidated pectin and a shell consisting of high mannuronic content alginate [21]. Such particles were able to move from aerogel to hydrogel, while absorbing high amounts of exudate (between 10 and 15 times their weight) and conforming to the wound cavity due to instantaneous collapse of the highly porous alginate shell (density 0.08 g/cm^3^) after contact with exudate. Fluid uptake as well as hydrogel formation rate into wound was controlled by the pectin core. Pectin as well as chitosan has been used alone or in combination with several polysaccharides to produce aerogel particles with proper wound healing properties. Chitosan aerogel particles with a high specific surface area (243–301 m^2^/g) and porosity (up to 98%) depending on chitosan molecular weight, had high hemostatic efficacy in vivo in a pig model over 3 h observation bleeding. Particles were able to adhere to the walls of the wound canal and formed a dense blood clot at the site of the femoral artery wound by sucking blood coming out from the artery [80].

Cellulose aerogel beads and particles have been prepared for wound healing by means of different techniques. Highly porous cellulose beads with bulk densities between 0.05–0.1 g/cm^3^ and specific surface areas between 200–500 m^2^/g can be prepared from cellulose–NaOH–water solutions [118]. The main parameters influencing the size and the shape of the beads were solution viscosity, and the strength of the shock of droplet on the surface of the coagulation bath. Highly spongy cellulose aerogel beads loaded with either hydrophilic or hydrophobic substances were prepared by the solvent releasing method and supercritical drying procedure (specific surface area of 371 m^2^/g) [119]. When the solvent removal was conducted under vacuum, the spongy cellulose structure collapsed, resulting in a dense matrix. Smaller cellulose aerogel particles with high specific surface areas (around 350 m^2^/g) were obtained by an emulsion technique via cellulose dissolution in NaOH-based solvent, non-solvent-induced phase separation, and drying with supercritical CO_2_ [120]. The size and specific surface area of the aerogel microparticles were strongly influenced by the coagulation phase taking place into the emulsion. In fact, the cellulose droplets may continue breaking up and/or coalescing with each other during mixing, leading to the formation of debris and then broad particle size distribution ranging between few tens of microns in diameter. Ultralight cellulose nanofibrils-based aerogel beads (20–50 μm in diameter) with bulk densities of about 0.0018 g/cm^3^ and very high porous structure were obtained by atomization [121]. Such particles had a specific surface area of about 390 m^2^/g and good porous structure that enabled good fluid uptake capacity. Covalent cross-linking between the native nanofibrils led to a stable 3D internal structure, enabling cell proliferation.

## 4. Bioaerogels Application Versus Wound Stages

In this section, different approaches to wound healing using aerogels at the different stages of the wound are described (Figure 4) and opportunities for the development of effective functional systems for therapeutic wound management are discussed.

### 4.1. Haemostasis

Effective hemostatic agents that rapidly control wound bleeding or trauma, together with the controlled release of drugs or biomolecules for wound healing, are critical for ensuring an adequate healing and avoiding bacterial infections, either in chronic or acute wounds. Certain types of materials, such as zeolites, mesoporous silica, or graphene sponges have been reported to promote the rapid absorption of the liquid component of the blood to accelerate the aggregation of red blood cells and platelets, and therefore, promoting hemostasis [122]. Others, such as those based on chitosan, are able to promote blood clotting due to its physicochemical/surface properties [123]. On the other hand, matrices such as gelatin, alginate and cellulose-based products, or combinations thereof, can incorporate bioactive agents to induce the blood coagulation pathway, such as, for instance, thrombin or fibrin [124]. The mechanism of action of these systems is based both on the ability to seal the wound and on the capacity to release these molecules, which favor hemostasis.

The potential of aerogels for wound healing applications is nowadays well demonstrated, taking into consideration their high fluid retention capacity, good mechanical properties and rapid shape recovery ability, generating the appropriate mechanical strength to reduce arterial blood loss. In addition, bioaerogel (pallets, particles and sponges) from several polymeric systems with remarkable hemostatic performance, inducing the platelet stimulation and stopping blood leakage, have been recently reported [69,125,126].

Fast expanding and highly absorptive compressed cotton-based aerogel pallets were developed for stopping blood leakage [69]. In particular, chitosan–cotton aerogels presented excellent properties to be used as hemostatic agents, especially in terms of higher volume expansion and reduced expansion time when compared with a commercial sponge. Chitosan, being positively charged, is able to attract negatively charged erythrocytes to form a layer of blood clot and, therefore, has been intensively studied in the field of hemostasis [123]. In another study, an injectable antibacterial aerogel with oxidized cellulose nanofiber and chitosan was developed for arterial blood loss in the face of penetrating or deep trauma [70]. These composite aerogels demonstrated to be a promising rapid hemostatic material for deep trauma, showing a good adhesion and aggregation effect to red blood cells and platelets, antibacterial activity and biocompatibility (Figure 5A). Chitosan has also been combined with diatom-biosilica to develop a composite aerogel with the potential to be a safe and rapid hemostatic material [127]. The chitosan/diatom-biosilica aerogel exhibited a multiscale hierarchical porous structure with favorable biocompatibility, efficient water absorption ratio and excellent hemostatic performance in vitro and in vivo (Figure 5B). The strong interface effect between the aerogel and blood was able to promote erythrocytes aggregation, platelets adhesion, and activation as well as to activate the intrinsic coagulation cascade to accelerate blood coagulation. More recently, the hemostatic behavior of chitosan aerogel particles were tested in vivo in a pig model, resulting in primary hemostasis and the formation of a dense clot inside the wound [80].

The potential use of crosslinked graphene sponge aerogels with hemostatic performance is gaining momentum due to the facile preparation, low cost, non-toxicity, and long shelf life. It was firstly reported by the group of Wang and collaborators [122], who produced ethanediamine crosslinked graphene aerogels, capable of absorbing blood plasma and allowing the accumulation of blood cells on its surface, which further promoted blood clotting on the wound (Figure 5C). The hemostatic process on the sponges’ surface mainly depended on the physical absorption process instead of clotting factors, as their surface did not stimulate hemocytes. The hemostatic potential of the graphene aerogels was improved by increasing the carboxyl groups at its surface through the crosslinking with 2,3-diaminopropionic acid (DapA), a medicinal amino acid [125], or with oxygen-containing groups by a mild cross-linking with polydopamine [126]. The obtained materials presented a high surface charge and strong platelet stimulation while exhibiting strengthened mechanical properties without compromising the absorbability, resulting in outstanding hemostatic performance (Figure 5D).

Using the same rationale of increasing the level of the negative charges at the surface of the material, stable composite aerogels from graphene oxide (GO) and polyvinyl alcohol (PVA), incorporating phenolic extracts from grape were prepared [129,130]. The developed aerogels exhibited the capacity for surface coagulation of whole blood, which was greatly improved by the incorporation and release of polyphenolic extracts with high content of proanthocyanidins, of potential use for wounds due to their wound healing and anti-inflammatory properties. Another bioactive natural compound, *Bletilla striata* polysaccharide, was combined with GO by hydrogen bonding to prepare a new composite aerogel sponge with hemostatic effect (Figure 5E) [128]. More recently, n-alkylated chitosan/graphene oxide porous sponges were developed for rapid and effective hemostasis in emergency situations [131]. Based on positive and negative charge attraction and hydrogen bonding interaction, a complex composite microstructure was generated with good absorption performance, mechanical strength and hemostatic efficacy.

Graphene-based aerogels have also been used as a matrix for eliminating heat injury of zeolite in hemostasis due to their thermal conductivity [132]. Zeolite is a well-known hemostatic agent, which generates heat upon application which can cause serious burns, if not properly controlled. The designed zeolite/cross-linked graphene sponge was able to manage the heat release of zeolite by the thermal conduction of graphene, while effectively ensuring blood clotting.

### 4.2. Inflammatory

Over the few last years, there has been an increasing interest in the application of biopolymer aerogels as anti-inflammatory matrices to improve the healing process. Several bioaerogels systems (scaffolds, particles, sensors and sponges) have been developed that try to respond to the problems that are imposed in wound treatment, essentially in the treatment and detection of infection. Several biopolymers (e.g., alginate, chitosan, cellulose and nanocellulose) can be used for producing aerogel matrices, as they can provide an optimal combination of properties such as low toxicity, biodegradability, high water uptake capacity, adhesive nature, minimum inflammatory response, enhanced stability upon storage, tunable drug release or even antimicrobial properties to prevent infections [84,89].

Aerogel porous 3D architectures are able to mimic the in vivo microenvironment, which endows them with greater water retaining capacity and flexibility, making them breathable for the transport of nutrients and gas in order to maintain a correct moisture balance at the wound site [81,95]. The porous structure also displays good thermal insulating properties to keep the wound temperature at a desired level [104]. Therefore, the interest of bio-based aerogels as wound dressings to address the inflammatory phase has grown and many works add the simultaneous ability of infection control [27]. On the other hand, aerogels can act as a controlled release system of therapeutic agents [31,73,88]. Therefore, the customization potential of aerogels as wound dressings, able to achieve different mechanical and physical properties, and bioactivity makes them exceptional materials for addressing wound healing at the inflammatory stage [71]. The aerogel-loading capacity of bioactive compounds, such as antibiotics, anti-inflammatory drugs and antioxidants (e.g., vancomycin [73], curcumin [95], doxycycline [21]) has been extensively studied. Several studies are reported in the present section; however, the combination of different drugs in the same aerogel system is not widely studied.

Upon absorbing the local fluids, aerogels can allow the formation of a wet gel into the wound site, keeping the proper balance of the exudate levels, and avoiding traumatic removal [23]. A wound dressing made of silk sericin and PVA was studied regarding the influence of the addition of glycerin in wound adhesion properties [133]. A full-thickness wound model of porcine skin indicated that the presence of glycerin decreased the cell adhesion while the in vivo tests on rats demonstrated the absence of tissue irritation and of severe signs of inflammation. These scaffolds with glycerin are less adhesive, which could reduce pain upon removal from the wound. In another study, a spongy wound dressing of pectin/carboxymethyl tamarind seed polysaccharide loaded with moxifloxacin beads for chronic wound healing was developed [134]. Moxifloxacin is a fourth-generation fluoroquinolone antibiotic and is beneficial for the treatment of infected wounds as it accelerates wound repair process [134]. The developed sponge possesses a wound exudate retaining ability that provides the ideal media to release the drug from beads and possesses degradable characteristics that reduce the frequency of wound dressing change. In vivo tests in rats demonstrated a severe reduction in inflammatory cells and the wound was totally closed, with the regeneration of hair follicles, after 17 days (Figure 6A). The spongy wound dressing has good healing and wound closing potential, achieved more contact with the wound area and has easy applicability.

Particulate systems with large surface areas and porosity are receiving attention in the field of wound treatments as drug delivery carriers, due to their capacity to control the inflammatory-reducing agents’ release on the wound site [81]. The influence of hybrid alginate aerogel particles and monoliths loaded with immune active compounds to accelerate wound treatment by targeting macrophage activity, excess-fluid absorption ability, antiseptic ability and controlled compound release were studied [72]. The presence of metal ions improves the wound healing, as Ca^2+^ ions are involved in inflammation and Zn^2+^ is vital for the immune system (phagocytosis, activity of polymorphonuclear leukocytes, maturation and activation of T- and B-lymphocytes). The presence of Ag^+^ in the aerogel increased the release of Zn^2+^. In other studies, chitosan aerogel beads loaded with vancomycin, an antibiotic, were prepared as a potential formulation to treat and prevent infections at the wound site after debridement [73,81]. The release of vancomycin depended on the particulate processing method, which occurred in 24 h using the jet cutting method, while when using the prilling method, the total drug release of drug was not observed in 7 days (Figure 6B). Both aerogel formats provide a fast release of vancomycin that allows the prevention of the infection, fully inhibiting the microbial growth of planktonic *Staphylococcus aureus* bacteria population in 48 h. Vancomycin–chitosan aerogel beads present high fluid sorption capacity, hemocompatibility, and cytocompatibility with BALB/3T3 mouse fibroblasts proving their effectivity in the prevention of wound site infections. In another approach, amidated pectin hosting doxycycline (broad range spectrum of action antibiotic drug) and alginate aerogel core–shell particles, by the prilling technique, were developed to be introduced into a gauze, slowing the drug release rate and improving the effectiveness of the drug on the wound site [21]. In the microcapsules production, alginate with high mannuronic content was used as a constituent of the shell and drug diffusion barrier, whereas amidated pectin loaded with doxycycline was used as the core of the particles (Figure 6C). The core–shell particles were able to prolong the release of doxycycline until 48 h, while the release of pure doxycycline was 20 min. The increase in the contact time of the drug with the wound allowed a higher control of inflammation. However, it is necessary to take into consideration the minimum dose desired to induce the treatment of inflammation. The majority of the papers do not address this issue, which we believe is critical to establish the lifetime of the aerogel dressing. Another important aspect is to recreate the wound environment in vitro in order to better predict the dressing performance in the real scenario.

Several published works are devoted to the production of hybrid or composite aerogels for wound healing using different combinations of natural polymers to decrease inflammation and prevent infection. Silk fibroin/chitin 3D nanocomposite scaffolds were developed as wound dressings, incorporating silver nanoparticles [137] and TiO_2_ nanoparticles [138]. Both aerogels presented the high absorption capacity of fluids, were biocompatible, biodegradable, had a blood clotting capability and high antimicrobial activity, which was increased by the presence of Ag and TiO_2_ nanoparticles, inhibiting the growth of *Escherichia coli*, *S. aureus*, and *C. albicans.* Recently, lightweight and mesoporous hybrid alginate–chitosan aerogel fibers were produced with demonstrated antimicrobial activity, against *S. aureus* and *K. pneumoniae,* and accelerated the wound closure in an in vitro (fibroblast) model for the single-layer recovery of damaged cells [84].

The bioactive functionalization of natural polymers was also carried out to improve aerogel properties. Curcumin cross-linked collagen aerogel scaffolds with controlled anti-proteolytic and pro-angiogenic activities exhibited good cytocompatibility and biochemical properties, and antimicrobial properties (*E. coli* and *Bacillus subtilis*), reducing the inflammatory response due to the beneficial healing properties of the curcumin [95]. Radwan-Pragłowska et al. [139] developed a novel microwave-assisted synthesis method of antioxidant chitosan aerogels containing *Tiliaplatyphyllos* extract, by freeze drying. *Tiliaplatyphyllos* have exceptional antioxidant and anti-inflammatory properties. Chitosan aerogel presented inhibition against *S. aureus*, non-cytotoxicity, antioxidant characteristics and a positive effect on fibroblasts cells proliferation. An alginate–chitosan interpolymer complex aerogel loaded with antibiotic levomycetin presented mesoporisty, a high surface area and a prolonged drug release (70% during 5 h) [140]. More recently, a Ca-Zn-Ag alginate aerogel for wound healing application was obtained by supercritical drying [135]. These aerogels presented an open-porous structure, high surface area, promoted rapid liquid uptake of human body fluids and were able to modulate inflammatory signaling pathways in macrophages showing high anti-inflammatory activity (Figure 6D). The anti-inflammatory activity of Zn-containing aerogels was evaluated in cultivated RAW 264.7 macrophages, using Toll-like receptor TLR4 as a mediator. TLR4 complexes are localized on macrophages’ surface and induce the production of pro-inflammatory molecules and can register the intact bacteria as well as isolated bacterial lipopolysaccharides [135]. Chitosan aerogels were prepared by dual crosslinking, chemical crosslinking by epichlorohydrin and physical crosslinking by itaconic acid to be used as a wound healing matrix [104]. The dual crosslinked structures improved the absorption ability by over 1200%, cell viability and structural stability without sacrificing the antibacterial properties. The high swelling ability of aerogels increased antibacterial properties.

Cellulose nanofibril (CNF) aerogel films were studied for the control of biofilm formation on chronic wounds [71,141]. A recent study demonstrated the clinical utility of CNF, with or without the incorporation of an antimicrobial alginate oligosaccharide (OligoG), to deliver antibacterial and antibiofilm agents on infection treatment [71]. The developed aerogel was able to modify bacterial growth and the biofilm development of *P. aeruginosa* and *S. aureus* and reduced bacterial virulence factor of *P. aeruginosa* production, supporting the potential of OligoG and CNF bionanocomposites for use in biomedical applications (Figure 6E) [71]. More recently, CNF aerogels were obtained by the supercritical impregnation of thymol, a monoterpene phenol found in essential oils and abundant in Thymus vulgaris with antibacterial, antifungal and anti-inflammatory activity [142]. For the first time, jackfruit and sugarcane aerogels silver decorated (Ag/JFA) with inherent structural anisotropy and antibacterial activity against *E. coli* and *S. aureus* was developed and incorporated into a 3D printable wearable bracelet intelligent wound management system (IWMS) [85]. The IWMS was produced with four incorporated functional units: Ag/JFA as the core material of a wearable bracelet; light-emitting diode; power supply system and 3D printing for a customized framework and circuit path. These ultra-light devices (including a power supply system) provided the highly efficient drainage of the exudate, LED wound warning (indicating by light color the warning role of an infected wound) and automatic enhanced antibacterial response.

Aerogels can be equipped with specific sensors that can report the presence of molecular compounds and detect changes in humidity and temperature at the wound site. In chronic wounds, the attention in protease detection is increasing, as proteolytic enzymes, in particular matrix metalloproteases (MMP) and serine protease human neutrophil elastase (HNE), responsible for the proteolytic degradation of growth factors and ECM, are present in the inflammatory phase [143]. Edwards et al. 2016 [143] developed a peptide–nanocellulose aerogels (PePNA) conjugate biosensor, made from unprocessed cotton and designed with protease detection activity. The PepNA detected the presence of HNE in chronic wounds, being a beneficial transducer surface as a biosensor layer for an intelligent protease sequestrant, suitable for interfacing with a wound dressing (Figure 6F). Following this study, cotton-based cellulosic filter paper [144], nanocellulosic crystals [144] and nanocellulosic aerogel [144] and peptide–cellulose conjugates prepared on a nanocellulosic aerogel [136] were evaluated as protease sensor materials in chronic wounds. Nanocellulosic materials had higher levels of peptide loading, surface charge, and protease sequestration compared to cellulosic materials, and improved sensitivity compared to the cellulose filter paper. The nanocellulosic biosensors detect HNE at concentrations found in chronic wounds. The development of such devices is essential to assist health professionals in the detection of infection, one of the problems raised in Section 1.

### 4.3. Proliferative

Aerogels present 3D, porous and ultralight structures which can be suitable for wound tissue regeneration. In fact, these structures can be similar to the physical characteristics of the natural ECM and may interact with adjacent cells, supporting cell attachment, spreading and proliferation [30]. Moreover, the outstanding properties of aerogels can allow for nutrient and oxygen supply to the cells and the elimination of cellular metabolic by-products, while supporting their homogeneous/isotropic growth towards tissue regeneration [22,30].

Several strategies for skin regeneration use aerogels integrated with bioactive molecules which are able to enhance the capability of active guidance and regeneration [40]. Several studies were developed using collagen, as it is the principal structural protein of ECM in connective tissues [30]. Hybrid collagen aerogel reinforced with wheat grass scaffolds were fabricated for collagen turnover and angiogenesis in wound healing applications [145]. Wheat grass is a bioactive molecule with potent anti-inflammatory, anti-bacterial, anti-fungal and antioxidant effects. This hybrid aerogel possesses a highly porous structure mimicking the natural ECM, promotes cell adhesion, growth and proliferation, and has antibacterial properties against *E. coli* and *B. subtilis.* The cumulative effect of the growth factors increased the angiogenic ability and collagen production of the aerogel by restoration of the damaged tissue, as noticed by the higher expression of collagen in fibroblasts and keratinocytes cells in in vivo tests (Figure 7A).

Biodegradable gelatin microspheres (GMs) containing basic fibroblast growth factor (bFGF) were incorporated into a porous collagen/cellulose nanocrystals (CNCs) scaffold by lyophilization [148]. The bioactivity of bFGF released were assessed by culturing HUVECs on scaffolds and it was demonstrated that the biological activity of bFGF was preserved. The scaffolds with bGFG-GMs promote HUVEC proliferation and viability. In vivo tests in a rat model showed the formation of blood vessels on the surface of collagen/CNCs/bFGF-GMs and collagen/CNCs/bFGF. After 14 days, the wound site was vascularized with the presence of fibroblasts, ECM formation and blood vessels formation. CD31 immunohistochemical staining of α-smooth muscle actin (α-SMA) was used as an angiogenesis biomarker, and a high density of mature (CD31 and α-SMA) blood vessels was formed in scaffolds loaded with bFGF. The developed scaffolds present high potential for wound regeneration.

Biocomposite aerogels of nanofibrillated cellulose (NFC) as matrix and copper-containing mesoporous bioactive glasses particles (Cu-MBG) as reinforced fillers were studied as wound dressing materials [149]. Cu^2+^ plays a key role in wound healing, reducing the risk of wound contamination and promoting wound repair directly by inducing angiogenesis and skin regeneration. A biological threshold of Cu^2+^ below 10 mg/L was suggested for the survival and growth of 3T3 fibroblasts, showing an angiogenic effect in HUVEC cells, by enhancing the gene expression of vascular endothelial growth factor (VEGF), PDGF and bFGF for 3T3 fibroblasts culture. The Cu^2+^ released also inhibited the growth of *E. coli*.

Chitosan/chondroitin sulfate (CS/ChS) aerogels were developed for the treatment of chronic skin lesions using a simple and low-cost production avoiding chemical reactions to minimize the risk of hypersensitivity reactions [150]. This material increased the granulation tissue and accelerated wound closure with pain suppression (Figure 7B) and presented angiogenic properties and good mechanical properties [146]. An electro assembly of chitin aerogel nanoparticles (CNPs) for wound healing was developed, showing a high water uptake ability and the promotion of wound healing on healthy male Sprague Dawley rats on day 9 [151]. Histological and immunohistochemical analyses revealed the formation of granulated tissue, deposition of collagen, generation of new blood vessels at day 7, whereas the total reepithelization and acceleration of the remodeling phase was observed at day 14. More recently, calcium alginate aerogels impregnated with mesoglycan were developed to favor the re-epithelialization and provide wound protection [102]. Mesoglycan has an antithrombotic and pro-fibrinolytic action in wound care [102]. In vitro tests confirmed the pro-migratory role of the produced aerogel once the migration rate of fibroblasts (BJ) and keratinocytes (HaCaT) cells increased in the presence of mesoglycan (Figure 7C). The developed aerogels assured a favorable and sterile environment for the wound healing process.

A biocompatible and ultra-lightweight graphene aerogel (GA) was fabricated by the one-step pyrolysis of glucose and ammonium chloride [147]. Normal lung epithelial cells were used to evaluate the biocompatibility via cell growth proliferation with efficient wound healing. The presence of GA increased ECM deposition and re-epithelization, inducing the expression of specific epithelial proteins, β-catenin and α-SMA. In addition, the GA inhibited the *S. aureus* growth (Figure 7D).

Overall, aerogels are able to recruit cells and generate signaling pathways towards angiogenesis, ECM formation and tissue regeneration, allowing the reduction of scar formation. The use of growth factors and other biologically relevant molecules can play an important role in the regeneration of viable and functional tissue. Nevertheless, the possibility of protein denaturation during the alcohol gel formation prior to scCO_2_ is a reality. Therefore, the combination of these bioactive molecules with biopolymer matrices that can have a protective function during aerogel fabrication is essential, opening room for many research possibilities [24,152].

Functional scaffolds have been formed by scCO_2_ incorporation biomolecules such as platelet lysates for the differentiation of human adipose-derived stem cells into an osteogenic lineage [153]. Using this strategy, the activity of the platelet lysates was preserved.

## 5. Conclusions and Future Perspectives

The treatment of wounds, in particular chronic and exudative, is an unsolved challenge due to its complex healing process, which often influences the quality of life of the patient. Although aerogels have just found their application as wound dressing materials, there have been important developments in the recent years that point out these outstanding structures as materials with great value for advanced wound healing and regeneration. Aerogel-based materials have been proposed from a variety of biopolymers to comprise a series of unique properties that are of utmost importance for wound healing applications. Namely, aerogels produced from natural-based polymers simultaneously offer low density, high porosity, high surface area, biocompatibility, biodegradability and the possibility to load and locally release therapeutic molecules at the wound site. Several technologies allow the fabrication of these different bioaerogel systems in the format of films, sponges and micro- and nanoparticle systems. This portfolio of aerogel processing possibilities generates advanced biopolymer nanostructured materials with applications in different wound scenarios, in particular those where fluids are continuously being produced and where its fluid control is a demand to progress to the more regenerative states. As hemostatic agents for deep trauma, several bioaerogels have presented a rapid blood clotting efficiency, such as cotton, chitosan, cellulose and graphene. Different aerogel structures were developed to unblock the inflammatory stage by combining the biological properties of biopolymers (silk fibroin, chitosan, collagen, alginate, pectin and cellulose) with the drug loading capacity of aerogels to accelerate wound treatment and prevent the infection. Due to their high surface area, aerogels can be equipped with specific sensors to report inflammatory signals such as the presence of neutrophil elastase or detect changes in humidity and temperature at the wound site. On the other hand, many of these systems are able to recruit cells and generate signaling pathways towards angiogenesis, ECM formation and tissue regeneration.

Bioaerogels face the challenge of degradation when in contact with the wound. It is mandatory to understand the degradation kinetics of the proposed systems to deeply study the interaction with the fluids (rates of absorption/retention) and to understand their degradation profile in realistic scenarios (simulated fluids). Only in this way is it possible to identify the optimal time window for the application of the materials in direct contact with the wound. Additionally, as medical devices, the need to fulfill sterilization requirements poses another important challenge, since conventional sterilization routes are not adequate to be applied for biopolymers. In this sense, supercritical CO_2_ can additionally bring value as a sterilization process for the bioaerogels presently under development [154,155]. Recently, supercritical technology has been used to sterilize and process biological tissue grafts [154,156,157]. Due to the mild working conditions, it is possible to preserve the properties of the tissue. In this way, the use of scCO_2_-based processing to produce aerogels using biological material can potentially give rise to a new generation of bioactive aerogel dressings.

Overall, substantial work already demonstrated that bioaerogels can allow for specific physical and biological interactions at the wound site, useful at a particular wound stage. Although aerogel is considered a high-cost technology, it is clear that by developing more functional wound dressings, it will be possible to decrease the time-to-heal of a wound, which in turn will result in an overall decrease in the treatment costs. This will result in an improvement of the patients’ life quality, reducing the time to return to active daily routines. The new horizons on clinical practice point towards personalized medicine strategies, as a way to provide more effective therapies and treatment protocols, which consequently will increase the success rates and improve the life quality of patients. In the context of wound healing, this becomes even more critical, since a wound is in constant transformation for the same patient. Therefore, this dynamic, variable and complex wound environment is the perfect scenario to challenge materials scientists, chemists and biomedical engineers to create new bioaerogel-based solutions for the development of advanced dressings able to promote not only the adequate wound environment, but also to explore the possibilities of wound-sensing, monitoring and responsiveness towards a therapeutic effect.

## Figures and Tables

**Figure 1 molecules-26-03834-f001:**
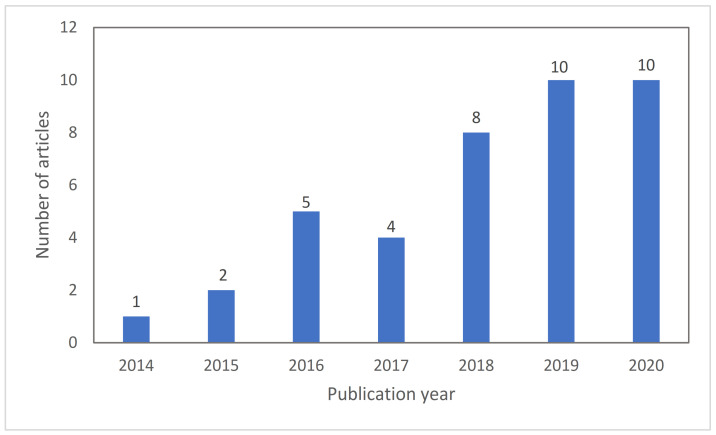
Evolution of scientific articles published in JCR-indexed journals on bio-based aerogels for wound treatment. Search criteria was “aerogel” AND “wound” in Web-of-Science database (source: WoS; search date: 2 February 2021), followed by the following refinements after hits revision: articles not intended for wound treatment or using porous materials that do not present mesoporosity were excluded, and addition of articles that were not in the database but were used for the preparation of this review article.

**Figure 2 molecules-26-03834-f002:**
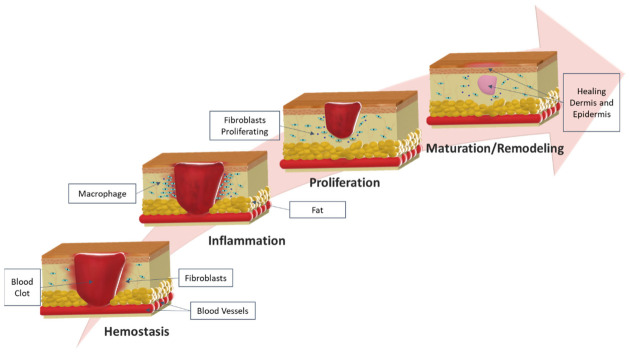
Schematic representation of wound healing stages (hemostasis, inflammation, proliferation and maturation/remodeling) and principal characteristics of each phase.

**Figure 3 molecules-26-03834-f003:**
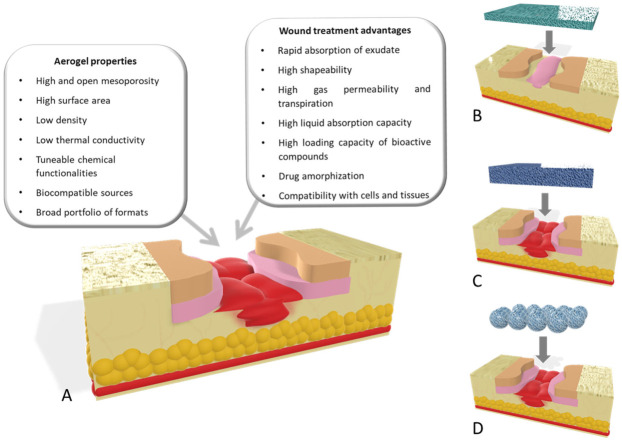
(**A**) Aerogel properties that are of utmost importance for wound healing. Different aerogel formats used: (**B**) aerogel films for wound healing applications in low-thickness wounds; (**C**) scaffolds; and (**D**) aerogel particles as wound healing applications for high-thickness wounds.

**Figure 4 molecules-26-03834-f004:**
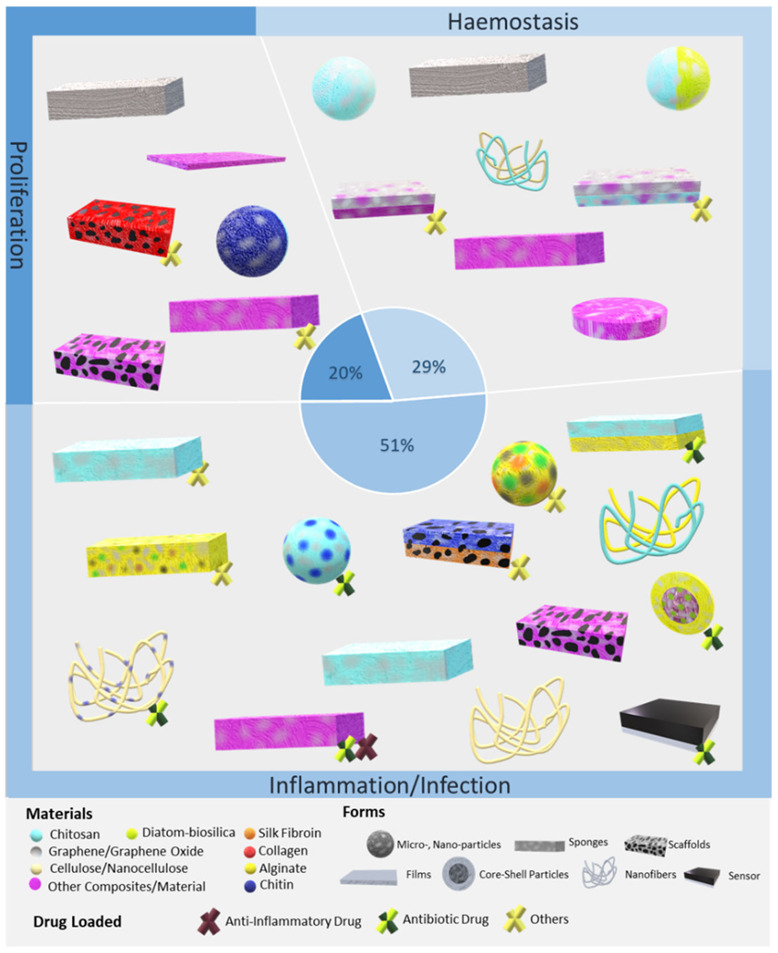
Bioaerogel formats with or without bioactive compounds that showed beneficial effects in the different phases of wound healing. Pie graph (center) denotes the percentage of articles on aerogels sorted by wound stage treatment. The same article database as in Figure 1 was used.

**Figure 5 molecules-26-03834-f005:**
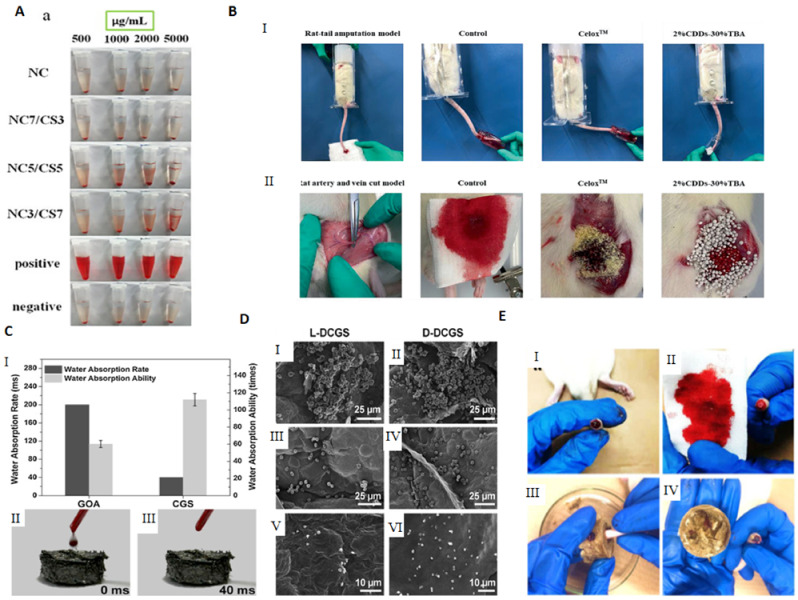
Use of bioaerogels in the treatment of wounds at the hemostasis stage: (**A**) photographs from hemolytic activity assay of the NC, NC7/CS3, NC5/CS5 and NC3/CS7 aerogels using sterile saline as negative control and deionized water as positive control. Notation: NC5/CS5 aerogel with 50 wt.% content of chitosan and cellulose nanofibers. Reprinted from ref [70], with permission from Elsevier. (**B**) In vivo hemostatic ability test and blood loss in the rat tail amputation model among the control, commercial product Celox, and chitosan/diatom-biosilica aerogel: (**I**) photographs of hemostatic effect and (**II**) blood loss in the rat tail amputation model. Adapted from ref [127] with permission from John Wiley and Sons. (**C**) (**I**) Water absorption rate and capability of cross-linked graphene sponge (CGS) and GO aerogel (GOA); (**II**) and (**III**) photographs of the blood absorption process on the CGS sample. Reprinted from ref [122], with permission from Elsevier. (**D**) (**I**,**II**) Images of cell morphology test of 2,3-Diaminopropionic acid cross-linked graphene sponge DCGS; (**III**,**IV**) hemocyte selective adhesion test; (**V**,**VI**) platelet selective adhesion test. Reprinted with permission from ref [125]. Copyright 2016 American Chemical Society. (**E**) Evaluation of the hemostatic performance of the bletilla striata polysaccharide/graphene oxide composite sponge (BGCS): (**I**) cutting the tail of the rat and bleeding; (**II**) 10 min application of gauze sponge resulted in continued bleeding; (**III**) BGCS was pressed on the wound; (**IV**) hemostasis was achieved by the BGCS. Reprinted from ref [128] with permission from Elsevier.

**Figure 6 molecules-26-03834-f006:**
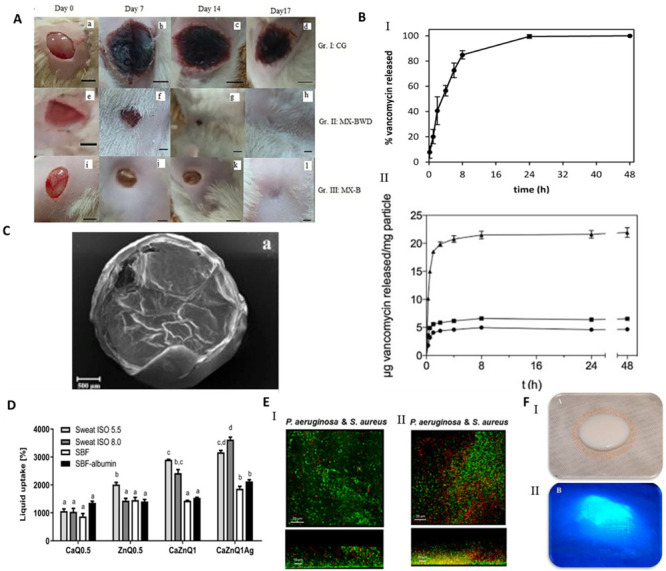
Use of bioaerogels in the treatment of wounds at the inflammatory stage: (**A**) In vivo wound healing (excision wound model) test in rat model consisting of five groups over a period of 17 days treatment. Group I: control (CG); Group II: moxifloxacin beads loaded in spongy wound dressing (MX-BWD) and Group III: moxifloxacin beads (MX-B). Adapted from ref [134], with permission from Elsevier. (**B**) Drug release of vancomycin from chitosan aerogels using sol–gel (**I**) jet cutting and (**II**) prilling methods. Reprinted from ref [81]. Reprinted from ref [73], with permission from Elsevier. (**C**) SEM image of core–shell alginate beads loaded with doxycycline showing the outer and inner structure of the beads. Reprinted from ref [21], with permission from Elsevier. (**D**) Liquid uptake quantification of alginate-based aerogels fluids incubated for 5 min in body fluid substitutes. Notation: CaQ0.5/ZnQ0.5 alginate aerogel with content of calcium/zinc. Alginate aerogel with content of calcium and zinc (CaZnQ1) and silver (CaZnQ1Ag). Reprinted from ref [135]. (**E**) LIVE/DEAD staining of 24 h dual-species biofilms grown in the presence of CNF aerogels. (**I**) Control of aerogels without OligoG (**II**) Oligo-bionanocomposites. Adapted from ref [71]. Copyright 2019 American Chemical Society. (**F**) pNA conjugate aerogels biosensor with protease activation with and without ultraviolet activation image (**I**,**II**): demonstrates the gauze contact layer and the aerogel biosensor layer with and without ultraviolet. Adapted from ref [136].

**Figure 7 molecules-26-03834-f007:**
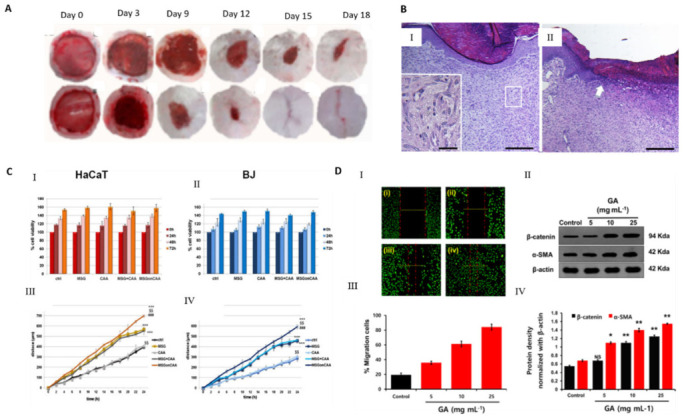
Use of bioaerogels in the treatment of wounds at the proliferative stage: (**A**) open excision wound healing test on native control collagen aerogels (first row) and on 2% wheat grass reinforced collagen aerogel (second row) in female Wistar rats. Adapted from ref [145]. Copyright 2017 American Chemical Society. (**B**) Micrographs of in vivo wound healing test showing (**I**) the appearance of the wound bed at day 14 after surgery. Treatment with CS/ChS aerogel has induced complete re-epidermization and proliferation of fibroblasts and production of collagen in dermis. The inset shows the dermis fibroblastic activity and neovascularization. An untreated control (**II**) shows at day 14 an epithelization tongue still in progress (white arrow) underneath the eschar (fibrin clot and necrotic detritus). In the dermis, scarring is observed with diffuse inflammatory infiltrated. Reprinted from ref [146], with permission from John Wiley and Sons. (**C**) MTT assays on (**I**) HaCaT and (**II**) BJ cells treated or not with pure mesoglycan (MSG), calcium alginate aerogel (CAA), MSG and CAA, and MSG impregnated on CAA, all of them at a final concentration of 0.3 mg/mL, at 24, 48 and 72 h; evaluation of migration rate of human (**III**) keratinocytes and (**IV**) fibroblasts in presence of pure MSG, CAA, MSG and CAA, and MSG impregnated on CAA. Reprinted from ref [102], with permission from Elsevier. (**D**) Cell proliferation on scratched lung epithelial cell surface promoted by GA: (**I**) cell growth migration in the control; (**II**) percentage index of migrated cells; (**III**) immunoblotting and (**IV**) intensity measurement of the expression levels of β-catenin and α-SMA on the scratched epithelial cell surface (5 × 10^4^ cells/well) before and after treatment with different concentrations of GA (* *p* ≤ 0.05; ** *p*≤ 0.01). Reprinted from ref [147]. Copyright 2019 American Chemical Society.

**Table 1 molecules-26-03834-t001:** Factors associated to Excessive or insufficient exudate production (adapted from [44,45,46]).

Examples of Factors	High Levels Exudation	Low Levels Exudation
Types of wounds	Chronic Venous Leg Ulcers Dehisced Surgical Wounds Malignant fungating wounds Burns Inflammatory ulcers—e.g., rheumatoid ulcers, pyoderma gangrenosum Skin donor sites	Ischemic/arterial wounds Neuropathic diabetic foot ulcers
Wound healing phase	Acute wounds—Beginning of healing process—inflammatory phase Chronic Wounds—prolonged inflammatory response	Towards end of healing process—proliferation and maturation phases
Wound size and depth	larger and deeper wounds	Smaller and superficial

**Table 2 molecules-26-03834-t002:** Types of wound exudate. Based on [44,55,56,57].

Type	Pictures	Color/Opacity	Consistency	Interpretations/observations
Serous	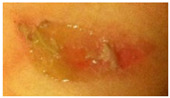	Clear, amber or straw-colored	Thin, watery	Normal during inflammatory and proliferative phases of healing Increase in in this type exudate may be a sign of infection or decompensation of comorbidities (e.g. Cardiac failure or malnutrition)
Serosanguineous	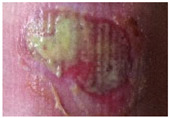	Clear, pink to light red	Thin, slightly thicker than water	May be considered normal during inflammatory and proliferative phases of healing. Pinkish (Rosy) due to the presence of red blood cells.
Sanguineous	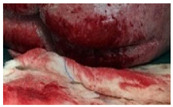	Red	Thin, watery	Reddish (Rose-colored) due to the presence of red blood cells. May indicate new blood vessel growth or the disruption of blood vessels or Capillary damage. Hypergranulation may be present.
Seropurulent	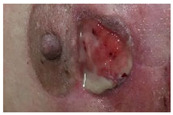	Cloudy, creamy, yellow, or tan	Thin	Serous exudate containing pus (*Presence of inflammatory cells, neutrophils and microorganisms*). May also be due to liquefying necrotic tissue. May signal impending infection.
Fibrinous	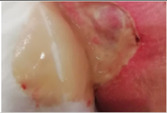	Cloudy	Thin, watery	May be associated to presence of fibrin strands. May indicate inflammation, with or without infection.
Purulent	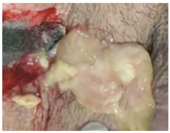	Opaque, milky, yellow, tan or brown, sometimes green	Viscous, sticky	Essentially pus Indicates infection. Specific coloration may be due to particular infection (e.g., Green coloration- *Pseudomonas aeruginosa*).
Haemopurulent	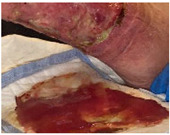	Reddish, milky, opaque	Viscous	Mixture of blood and pus. Often due to established infection.
Hemorrhagic	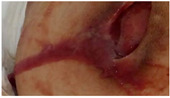	Dark red	Viscous	May be associated with Trauma. Capillary Friability. May indicate bacterial infection.

**Table 3 molecules-26-03834-t003:** Dressing-exudate interaction [44,45,59].

Status	Pictures	Indicators	Comments
Dry	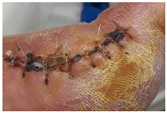	Wound bed is dry; there is no visible moisture and the primary dressing is unmarked; Not an ideal wound healing environment;	Dressing may be adherent to wound. Dressing material with no signs of exudation Dry perilesional skin
Moist	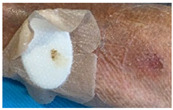	Small amounts of fluid are visible when the dressing is removed; wound bed could appear glossy;	Good exudate management—dressing change frequency is appropriate for dressing type; Conditions surrounding skin may be intact and hydrated.
Wet	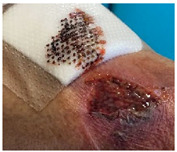	Small amounts of fluid are visible when the dressing is removed; the primary dressing may have absorbed large amounts of exudation	Apparent imbalance in exudate management—Dressing material passed through but not saturated. Dressing material has absorbed large amounts of exudate, assess frequency of dressing change which may be appropriate Potential for peri wound maceration
Saturated	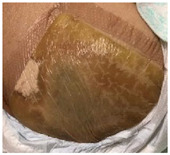	Primary dressing is wet and may be saturated; Not an ideal wound healing environment;	Imbalance in exudate management. It is necessary to simultaneously adjust the frequency of dressing change (dressing change is required more frequently than usual for the dressing type). Maceration and loss of perilesional skin integrity may be present.
Leaking	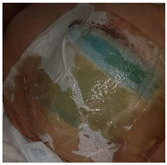	Dressings are saturated and exudate is escaping from primary and secondary dressings onto clothes or beyond; Not an ideal wound healing environment;	Mismanagement of exudate. Possibility of material degradation. Dressing change is required much more frequently than usual for dressing type. Macerated surrounding skin with loss of skin integrity

**Table 4 molecules-26-03834-t004:** Key factors affecting dressing/device choice. Adapted from [44].

Clinical Need and Indications for Use
Type of Wound
Size/depth
Location, position and shape of the anatomical area
Tissue type
Exudate Amount
Antimicrobial indication
Need of odor control
Allergies/sensitivities to dressing materials or contents
Availability/inclusion in local formulary
Reimbursement policies
Dressing/device change frequency Clinician preference/experience Economic capacity of the Patient and particular preferences

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
