# Peer review of "Bioaerogels: Promising Nanostructured Materials in Fluid Management, Healing and Regeneration of Wounds"

_molecules, 2021, doi:10.3390/molecules26133834_

Round 1

Reviewer 1 Report

Manuscript ID molecules-1230153   Bioaerogels: promising nanostructured materials in fluid management, healing and regeneration of wounds is well done and could be suitable for publication in the fom it is present.

Author Response

Comments and Suggestions for Authors   Manuscript ID molecules-1230153   Bioaerogels: promising nanostructured materials in fluid management, healing and regeneration of wounds is well done and could be suitable for publication in the fom it is present.  

Response: The authors would like to begin by thanking the reviewer for this opinion and  consideration. We are thankfully for your interest and excellent evaluation of our work entitled “Bioaerogels: promising nanostructured materials in fluid management, healing and regeneration of wounds”.

Reviewer 2 Report

This is an interesting overview of bioaerogels. I suggest that the Authors explain more fully the potential different role of bioaerogels in traumatic and surgical wounds, because they have some relevant differences. On surgical wounds, I suggest that the Authors consider a recent manuscript regarding wound healing in urological surgery that may complement your treatment.(PMID:32545258; PMCID: PMC7356923DOI: 10.3390/jcm9061822)

I also suggest reducing each part of the manuscript, which in its present form is too much like a book chapter rather than an article for a scientific journal.

Author Response

Comments and Suggestions for Authors

This is an interesting overview of bioaerogels. I suggest that the Authors explain more fully the potential different role of bioaerogels in traumatic and surgical wounds, because they have some relevant differences. On surgical wounds, I suggest that the Authors consider a recent manuscript regarding wound healing in urological surgery that may complement your treatment.(PMID:32545258; PMCID: PMC7356923DOI: 10.3390/jcm9061822)

Response: The authors would like to thank your comments on the treatment and healing of surgical wound. The authors would like to inform that your comment allows us to improve our introduction having in consideration the surgical wounds (Page 2, line 65/66). In relation to the mentioned literature regarding silver nanoparticles, this kind of particles have been highlighted in section 4.2 (Page 24, line 814/815) in the context of aerogel based systems.

I also suggest reducing each part of the manuscript, which in its present form is too much like a book chapter rather than an article for a scientific journal.

Response: The authors would like to thank your comment and suggestion. We proceed to the reducing of each part of the manuscript, focusing on the essential aspect of the article, application of Bioaerogels in wound treatment.

Reviewer 3 Report

Journal: molecules

Manuscript ID: molecules-392668

Article Type: Original Research Article

Title: Bioaerogels: promising nanostructured materials in fluid man-agement, healing and regeneration of wounds

Authors: Beatriz G. Bernardes, Pasquale del Gaudio, Paulo Alves, Raquel Costa, Carlos A. García-Gonzaléz and Ana Leite Oliveira1

Keywords: Aerogels; Biopolymer; Exudate; Wound Healing

This paper titled, " Bioaerogels: promising nanostructured materials in fluid man-agement, healing and regeneration of wounds " is a review manuscript. Combining the characteristics of aerogels with fluid interaction and drug loading, advanced biopolymer nanostructure materials have been developed, and they have been applied in different wound conditions. Eliminate the inflammatory phase, thereby speeding up wound treatment and preventing infection. To stop bleeding and promote the healing and regeneration of exudative and chronic wounds. In fact, aerogel sponges from several polymer systems with excellent hemostatic properties have recently been reported. The composite aerogel has been proven to be a promising rapid hemostatic material for deep wounds. It has good adhesion and aggregation effects on red blood cells and platelets, can load and locally release therapeutic molecules on the wound site, and has antibacterial activity and biological characteristics. However, this work is not critical and presents recent breakthrough in the scope of Molecules.

Error

  1. In paragraph 3 of Introduction, the sum of percentages is not one hundred percentage. Why?
  2. In Table 3, Type: Sanguineous Comments, blook cells.→ blood cells.

Sum up, a review article should conduct a scientific analysis of the examples, and put forward authors’ own views and practical suggestions based on the literature. This article only is a listing of the results of literatures, not suitable for our journal, Molecules.

Author Response

Comments and Suggestions for Authors

Journal: molecules

Manuscript ID: molecules-392668

Article Type: Original Research Article

Title: Bioaerogels: promising nanostructured materials in fluid man-agement, healing and regeneration of wounds

Authors: Beatriz G. Bernardes, Pasquale del Gaudio, Paulo Alves, Raquel Costa, Carlos A. García-Gonzaléz and Ana Leite Oliveira1

Keywords: Aerogels; Biopolymer; Exudate; Wound Healing

This paper titled, " Bioaerogels: promising nanostructured materials in fluid man-agement, healing and regeneration of wounds " is a review manuscript. Combining the characteristics of aerogels with fluid interaction and drug loading, advanced biopolymer nanostructure materials have been developed, and they have been applied in different wound conditions. Eliminate the inflammatory phase, thereby speeding up wound treatment and preventing infection. To stop bleeding and promote the healing and regeneration of exudative and chronic wounds. In fact, aerogel sponges from several polymer systems with excellent hemostatic properties have recently been reported. The composite aerogel has been proven to be a promising rapid hemostatic material for deep wounds. It has good adhesion and aggregation effects on red blood cells and platelets, can load and locally release therapeutic molecules on the wound site, and has antibacterial activity and biological characteristics. However, this work is not critical and presents recent breakthrough in the scope of Molecules.

Response: The authors are very grateful for the careful revision of this manuscript, positive feedback and relevant comments.

Error

In paragraph 3 of Introduction, the sum of percentages is not one hundred percentage. Why?

In Table 3, Type: Sanguineous Comments, blook cells.→ blood cells.

Response: The authors are very grateful for the alert to errors (in paragraph 3 and table 3), which were all taken into consideration and have been corrected and highlighted.

Sum up, a review article should conduct a scientific analysis of the examples, and put forward authors’ own views and practical suggestions based on the literature. This article only is a listing of the results of literatures, not suitable for our journal, Molecules.

Response: According to the suggestion, the authors tried to improve discussion at the different sections of manuscript, trying to introduce critical analysis and to evaluate the strong and weak aspects of the different developed systems.

Reviewer 4 Report

Bio-based aerogels are particularly attractive since they encompass theirintrinsic chemical properties and the physical features of their nanostructure. The emerging research on aerogels for wound treatment is reviewed in this manuscript. The scenario and the opportunities provided by aerogels in the form of films, membranes and particles are identified to face demands in fluid managing and wound healing and regeneration.

It is recommended to focus on aerogels in the second part (“Challenges and opportunities to heal exudative wounds”), so I think it can be simplified in this part.

Author Response

Comments and Suggestions for Authors

Bio-based aerogels are particularly attractive since they encompass theirintrinsic chemical properties and the physical features of their nanostructure. The emerging research on aerogels for wound treatment is reviewed in this manuscript. The scenario and the opportunities provided by aerogels in the form of films, membranes and particles are identified to face demands in fluid managing and wound healing and regeneration.

It is recommended to focus on aerogels in the second part (“Challenges and opportunities to heal exudative wounds”), so I think it can be simplified in this part.

Response: The authors are very grateful for the careful revision of this manuscript, positive feedback and relevant comments. After a critical analysis, the authors agreed with your suggestions, simplifying the section 2 of the review article “Challenges and opportunities to heal exudative wounds”. This section is now more clarify and detailed in order to focus on the relevant aspects.

Round 2

Reviewer 2 Report

The Authors have only partially followed my suggestions. The manuscript is still too redundant. Furthermore, the Authors did not add in the references the manuscript I suggested regarding the clinical relevance of nanoparticles. I do not believe that the authors have responded adequately and have followed my indications. I cannot recommend the manuscript in its present form for publication.

Reviewer 3 Report

accept